# Enhanced recovery programmes versus conventional care in bariatric surgery: A systematic literature review and meta-analysis

**Khalid Al-Rubeaan[1], Cindy Tong[2], Hannah Taylor[3]\*, Karl Miller[4], Thao Nguyen Phan Thanh[5], Christian Ridley[6], Sara Steeves[6], William Marsh[6]**

**1** Research and Scientific Center, Sultan Bin Abdulaziz Humanitarian City, Riyadh, Kingdom of Saudi Arabia, **2** Johnson & Johnson Medical Devices, Somerville, New Jersey, United States of America, **3** Johnson & Johnson Medical Limited, Leeds, United Kingdom, **4** Johnson & Johnson Middle East, FZ LLC, Dubai, United Arab Emirate, **5** Johnson and Johnson Medical SAS, Saint-Priest, France, **6** Costello Medical Consulting Limited, Cambridge, United Kingdom

\* htaylor8@its.jnj.com

**Data Availability Statement:** All relevant data are within the paper and its Supporting Information files.

## Abstract

### Background

With obesity prevalence projected to increase, the demand for bariatric surgery will consequently rise. Enhanced recovery programmes aim for improved recovery, earlier discharge, and more efficient use of resources following surgery. This systematic literature review aimed to evaluate the evidence available on the effects of enhanced recovery programmes after three common bariatric procedures: laparoscopic Roux-en-Y gastric bypass (LRYGB), laparoscopic sleeve gastrectomy (LSG), and one anastomosis gastric bypass (OAGB).

### Methods

MEDLINE, Embase, the Cochrane Library and the National Health Service Economic Evaluation Database were searched for studies published in 2012–2019 comparing outcomes with enhanced recovery programmes versus conventional care after bariatric surgery in Europe, the Middle East and Africa. Data were extracted and meta-analyses or descriptive analyses performed when appropriate using R.

### Results

Of 1152 screened articles, seven relevant studies including 3592 patients were identified. Six reported outcomes for 1434 patients undergoing LRYGB; however, as only individual studies reported on LSG and OAGB these could not be included in the analyses. The meta-analysis revealed a significantly shorter mean duration of hospital-stay for LRYGB enhanced recovery programmes than conventional care (mean difference [95% CI]: -1.34 days [-2.01, -0.67]; p<0.0001), supported by sensitivity analysis excluding retrospective studies. Meta-analysis found no significant difference in 30-day readmission rate (risk ratio [95% CI]: 1.39 [0.84, 2.28]; p = 0.2010). Complication rates were inconsistently reported by

**Funding:** This study was sponsored by Johnson & Johnson Medical NV. CT, HT, KM and TN are employees of Johnson & Johnson. KA, CR, SS and WM served as paid consultants for Johnson & Johnson. The funder provided support in the form of salaries for CT, HT, KM and TN, remuneration for KA, CR, SS and WM, and contributed to the study design, data collection and analysis, decision to publish, and preparation of the manuscript. The specific roles of these authors are articulated in the 'author contributions' section. All authors had full access to the data, reviewed and approved the final version, and were responsible for the decision to submit for publication. A medical writing agency, employed by the funder, assisted with manuscript preparation under the authors' direction.

**Competing interests:** I have read the journal's policy and the authors of this manuscript have the following competing interests: CT, HT, KM and TN: Employees of Johnson & Johnson KA: Employee of Sultan Bin Abdulaziz Humanitarian City at the time this study was conducted, and paid consultant for Johnson & Johnson CR, SS and WM: Employees of Costello Medical at the time this study was conducted and have served as paid consultants for Johnson & Johnson These affiliations do not alter our adherence to PLOS ONE policies on sharing data and materials.

Clavien-Dindo grade, but descriptive analysis showed generally higher low-grade rates for enhanced recovery programmes; the trend reversed for high-grade complications. Reoperation rates were rarely reported; no significant differences were seen.

## Conclusion

These results support enhanced recovery programmes allowing shorter inpatient stay without significant differences in readmission rate following LRYGB, although complication and reoperation rate comparisons were inconclusive. Further research is needed to fill current data gaps including the lack of studies on LSG and OAGB.

## Introduction

Bariatric surgery is recommended as an intervention for obesity and type 2 diabetes mellitus (T2DM) by a number of societies, including the American Diabetes Association, Diabetes UK and the Obesity Management Task Force of the European Association for the Study of Obesity, and is recommended in a joint statement by international obesity organisations from the 2nd Diabetes Surgery Summit in 2016 [1–4]. The World Health Organization estimated approximately 20% of adults in the Middle East and Africa, and 23% of adults in Europe had a body mass index (BMI) of 30 or higher in 2016 [5]. Severe obesity can increase the risk of hypertension, hyperlipidaemia, heart disease and ischaemic stroke, as well as a number of cancers including cancer of the colon, gall bladder, rectum and liver [6–10]. Obesity is also closely linked to the development of T2DM due to a progressive decrease in insulin secretion alongside a rise in insulin resistance [11,12]. Obesity is estimated to account for 65–80% of T2DM cases in Europe, and a 2016 study estimated that 48% of obese males and 77% of obese females in Kuwait were also diabetics [13,14].

The National Institute for Health and Care Excellence (NICE) has recognised that bariatric surgery is both clinically useful and cost effective for patients with recent onset T2DM who are also obese with a BMI of 35–40 kg/m$^2$ [15]. Bariatric surgery can lead to significant, sustained weight loss in obese patients and increased glycaemic control in patients with uncontrolled T2DM [16–18]. It has also been shown to improve insulin resistance [17].

Although the average cost of individual bariatric surgery in the United Kingdom is reported to be £9164, it has been found to reduce the overall healthcare costs over a patient's lifetime [19,20]. The economic benefits further extend to societal savings through increased workplace productivity [21,22]. However, the rate of bariatric surgery utilisation varies significantly across Europe, where less than 1% of those in the United Kingdom who could benefit from bariatric surgery receive treatment [23,24].

The most common bariatric surgery procedures are laparoscopic Roux-en-Y gastric bypass (LRYGB), laparoscopic sleeve gastrectomy (LSG) and one anastomosis gastric bypass (OAGB) [25]. Enhanced recovery programmes (ERPs) comprise pre-, intra- and postoperative measures that aim to minimise patients' physiological stress response to surgery, lower the incidence of complications, allow earlier discharge and use hospital resources more efficiently [26]. These measures include recommendations for patients to stop smoking and lose weight prior to surgery, a laparoscopic approach, postoperative monitoring of the patient's protein intake and monitoring of the frequency of apnoeic episodes [26]. ERPs have the potential to decrease the cost and improve surgery efficiency without compromising clinical outcomes or increasing the risks of reoperation and readmission [27].

While previous meta-analyses have compared outcomes between ERPs and conventional care in bariatric surgery, the authors of these analyses have noted the limited evidence base that informed them [28,29]. The literature searches to inform these analyses were conducted in 2016 and 2017, respectively, and with the recent publication of additional comparative studies this study sets out to evaluate what evidence gaps remain around the use of ERPs for LRYGB, LSG and OAGB [30–32]. A systematic literature review (SLR) was therefore carried out to identify recent clinical studies and clinical practice guidelines describing the use of any ERP in comparison with conventional care for patients undergoing these procedures across Europe, the Middle East and Africa (EMEA). Patient and physician-reported clinical, safety and economic outcomes were targeted for the review, and where possible, meta-analyses were conducted to compare outcomes between ERPs and conventional care for each procedure type.

## Materials and methods

An SLR was carried out following a pre-specified protocol in accordance with the PRISMA statement (S1 Table), initially in October 2017 for clinical, patient-reported and economic outcomes associated with ERPs following bariatric surgery in EMEA and later updated in July 2019 for studies comparing ERPs with conventional care in LRYGB, LSG and OAGB specifically [33]. This manuscript incorporates the comparative studies obtained across both reviews.

MEDLINE, Embase, the Cochrane Library (Cochrane Database of Systematic Reviews, Database of Abstracts of Reviews of Effect, Cochrane Central Register of Controlled Trials) and the National Health Service Economic Evaluation Database were searched on 18th October 2017, and again on 15th July 2019 in the update. The search terms used are provided in S2–S5 Tables. Grey literature sources were searched including abstract books of major surgical and economic congresses between 2015 and 2019 and expert recommendations from the websites of clinical societies and organisations (S6 Table). This review excluded SLRs and meta-analyses. However, supplementary hand searches were conducted to identify any studies included in relevant SLRs and meta-analyses that were not identified in the electronic database searches.

Two independent reviewers assessed the titles and abstracts of all search results (stage 1), as well as the full texts of all potentially eligible studies identified in stage 1 (stage 2). A third independent reviewer resolved any disagreements. Eligible publications included studies in patients undergoing LRYGB, LSG or OAGB that had implemented an ERP, defined in this review as programmes with a multi-component, perioperative protocol that focused on optimising patient recovery and discharge. Studies had to report at least one of the following outcomes: guidelines and recommendations from a formal clinical society; efficacy, safety or tolerability outcomes; quality of life and other patient-reported outcomes; costs and resource use relating to the ERP; and to have compared outcomes with conventional care. As the way ERPs are implemented varies in different regions, this review focused specifically on data from the EMEA region, requiring at least some of the patients in each study to be within this region. Similarly, only articles published in or after 2012 were included in the review to focus on ERPs based on more current definitions of ERPs used in clinical practice. Finally, studies were required to be published in a European language, to report outcomes separately for LRYGB, LSG and OAGB, and, in order to avoid studies with very low patient numbers available for analysis, to have at least 30 patients in a given surgical arm. Detailed eligibility criteria are given in S7 Table.

Data from all included studies were extracted into pre-specified extraction grids in Microsoft Excel (S8 Table). Data were extracted by one reviewer and a second reviewer independently verified the extracted information. Discrepancies were resolved by discussion until a

consensus was reached or, where necessary, a third independent reviewer made a final decision.

All extracted articles underwent a quality assessment for risk of bias by a single reviewer. A second reviewer independently verified the quality assessment, with discrepancies arbitrated by a third individual. Quality assessments of randomised controlled trials (RCTs) were based on the template provided in the NICE single technology appraisal manufacturer's template [34]. For non-randomised studies, a modified version of the Downs and Black checklist was used in which questions that were only relevant for randomised studies or inappropriate for the assessed studies were removed (S9 Table) [35].

After extractions were completed, outcomes reported in the selected studies were reviewed for suitability for inclusion in a meta-analysis. There was substantial variation in how outcomes were collected and reported in these studies, and only analyses of inpatient length of stay in hospital and 30-day readmission rate were ultimately considered feasible for meta-analysis. Descriptive analyses were conducted for 30-day reoperation and complication rates, as the next most reported outcomes.

Meta-analyses were conducted in R using random effects models with a maximum likelihood estimator. The median was used to estimate mean values for studies where length of stay was only reported as a median and range, and, as the sample size was greater than 70 for each of these studies, range/6 was used to estimate standard deviation [30,36,37]. Sensitivity analysis excluding these studies produced the same result as the primary analysis (S1 Fig). A maximum likelihood estimator was chosen as simulation studies have demonstrated this to have suitable properties for estimating between-study variance [38,39]. Heterogeneity was assessed with Cochran's Q and $I^2$ statistics. The primary analyses were conducted with all relevant studies, while sensitivity analyses excluding retrospective studies were conducted to assess the impact of variation in study design on the meta-analysis outcomes.

## Results

### Study selection

Overall, 1014 articles from database searches and 138 articles from hand searches were screened for relevance. Following article screening, six studies reporting relevant outcomes on the use of ERPs versus conventional care for LRYGB were included (Fig 1) [30–32,36,40,41]. Only individual studies were identified for LSG and OAGB, therefore meta-analysis was not possible for these procedures. Data from these studies are provided in S10 Table.

### Study and patient characteristics

Table 1 summarises the characteristics of the LRYGB studies included in the review. Two studies were RCTs, two were prospective observational studies, and two were retrospective observational studies. The mean age of included participants ranged from 36.1 to 46.2 and 38.1 to 48.4 for the conventional care and ERP arms, respectively. Mean BMI ranged from 44.5 to 46.8 in the conventional care and 42.8 to 44.9 in the ERP arms, where reported. The prevalence of comorbidities varied across the studies with higher rates of sleep apnoea and dyslipidaemia reported by Ruiz-Tovar et al. (2019) [32].

ERP implementation varied somewhat across studies, with the more recent European studies showing the closest alignment with Enhanced Recovery After Surgery (ERAS®) Society guidelines (Fig 2) [26]. However, it was not possible to determine from the publications if or how a large number of the ERAS® guidance elements had been implemented, making it difficult to assess the level of variation between studies and potentially limiting the comparability of the studies' outcomes.

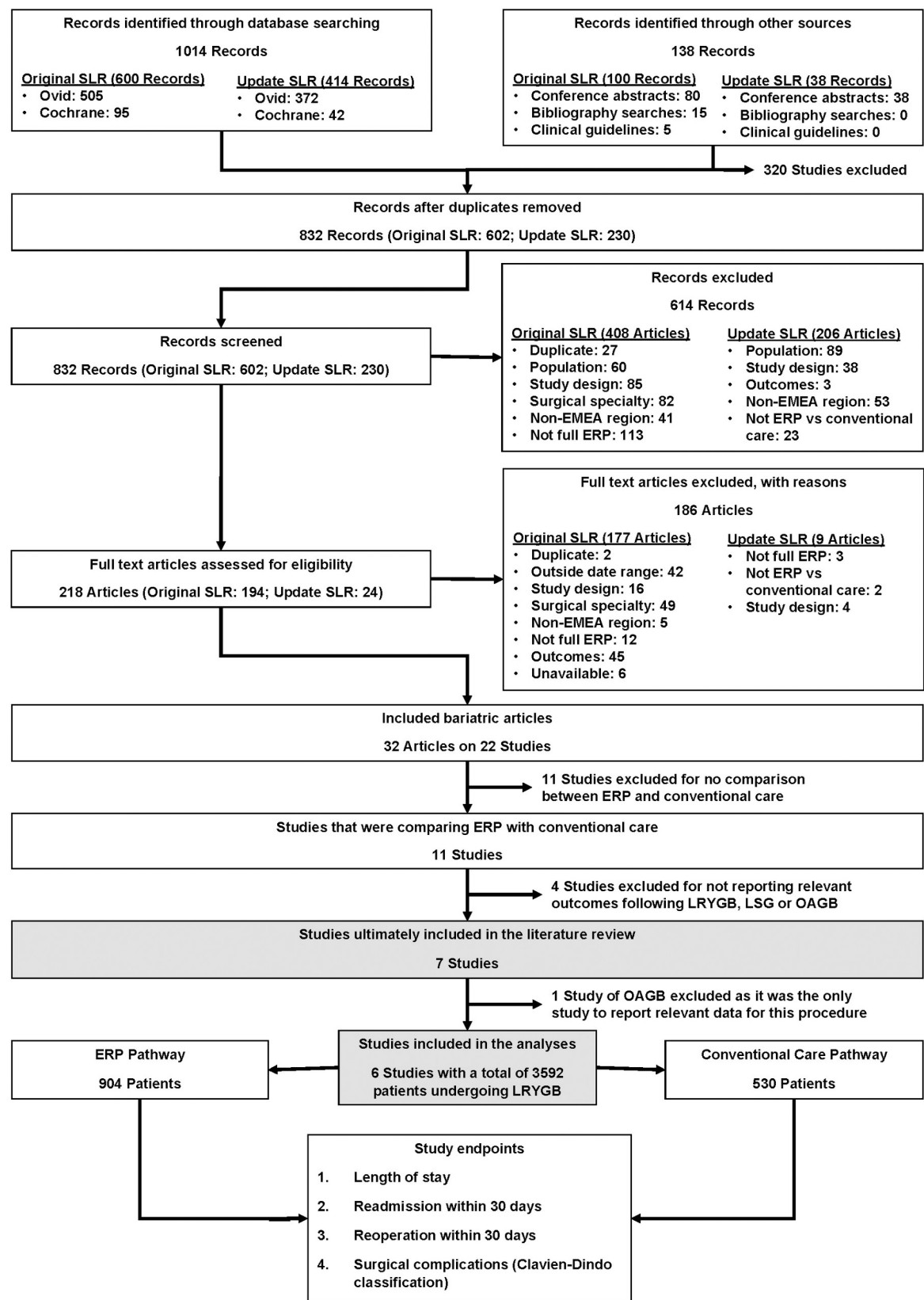

**Fig 1. PRISMA diagram of study selection.** EMEA: Europe, the Middle East and Africa; ERP: enhanced recovery programme; LRYGB: Laparoscopic Roux-en-Y gastric bypass; LSG: Laparoscopic sleeve gastrectomy; OAGB: One anastomosis gastric bypass; SLR: Systematic literature review.

Most articles demonstrated a low risk of bias according to quality assessment (S11 Table). Areas with the highest risk of bias included inadequate concealment and blinding of treatment allocation in the RCTs, poor description of patients lost to follow-up and not recruiting patients from ERP and conventional care arms over the same period in the observational studies. Funnel plot assessment for publication bias was not conducted due to the small number of included studies [42].

### Length of stay

Hospital length of stay was reported by all six included studies. From the meta-analysis, patients receiving ERP care had a shorter length of stay compared to those receiving conventional care, with a statistically significant mean difference of -1.34 days (p<0.0001; 95% CI: -2.01, -0.67) for ERPs versus conventional care (Fig 3). This demonstrated that ERPs led to significantly shorter lengths of stay than conventional care after LRYGB surgery, however, high heterogeneity was seen between the studies ($I^2$ = 97.1%). The sensitivity analysis excluding retrospective studies (S2 Fig) supported this conclusion, with a mean difference of -1.09 days (p<0.0001; 95% CI: -1.95, -0.22); however, a high level of heterogeneity remained ($I^2$ = 98.3%).

### Readmission rate

All six LRYGB studies reported 30-day readmission rate data. Meta-analysis of the risk ratio for readmission when receiving ERP versus conventional care was performed, finding no significant difference (risk ratio: 1.39 [p = 0.2010; 95% CI: 0.84, 2.28]; Fig 4). A sensitivity analysis was performed, excluding retrospective studies (S3 Fig). The risk ratio was 1.18 (95% CI: 0.64, 2.18), and remained not statistically significant (p = 0.5859). Similar results were seen for each study individually.

### Complications

Three studies reported 30-day complication rates by Clavien-Dindo grade in conventional care versus ERP arms (Fig 5 and Table 1) [30,31,36]. Only two studies reported Grade IVA/IVB rates, and one reported Grade V rates. When the complication rates were aggregated across the included studies, the aggregated rate was somewhat higher at lower grades for ERP than for conventional care (Grade I: 9.6% [61/636] ERP vs 8.0% [21/262] conventional care; Grade II: 12.7% [81/636] vs 7.3% [19/262]) but this trend was reversed at higher grades (Grade IVA: 1.5% [8/526] vs 2.0% [3/152]).

Two studies reported rates of pneumonia, finding no significant differences between ERP and conventional care groups; Geubbels et al. (2014) reported 1.7% (6/360) of patients in the ERP and 2.9% (3/104) in the conventional care group developed pneumonia while Mannaerts et al. (2019) reported no cases in either group [31,36]. Surgical site infection rates were similarly low and not significantly different between ERP and conventional care groups [31,36]. Only Mannaerts et al. (2019) reported rates of deep vein thrombosis (0.6% [1/166] in the ERP group and 0% in the conventional care group), and no studies reported pulmonary embolism rates [31].

**Table 1. Summary of demographic and study characteristics in addition to complication and reoperation rates.**

| Study author, country and year | Type of study | Type of care | Number | Mean age (SD) | Male n (%) | Mean BMI (SD) | Comorbidities n (%) | | | | 30-day reoperations n (%) | 30-day Clavien-Dindo grade complications n (%) | | | | | | |
| | | | | | | | Diabetes | Hypertension | Dyslipidaemia | Sleep apnoea | | Grade I | Grade II | Grade IIIA | Grade IIIB | Grade IVA | Grade IVB | Grade V |
|---|---|---|---|---|---|---|---|---|---|---|---|---|---|---|---|---|---|---|
| Geubbels, N. Netherlands, 2014 | Retrospective | ERP | 360 | 41.7 (10.1) | NR (19.2) | 42.1 (34.5–67.6)[a] | NR (24.4)[b] | NR 40.8 | NR (26.2) | NR (11.7) | NR | 26 (7.2) | 27 (7.5) | 2 (0.6) | 4 (1.1) | 7 (1.9) | 0 | 0 |
| | | CC | 104 | 42.8 (10.2) | NR (23.1) | 43.2 (35–61.5)[a] | NR (34.6)[b] | NR (44.2) | NR (24.3) | NR (18.3) | NR | 1 (1) | 10 (9.6) | 1 (1) | 4 (3.8) | 2 (1.9) | 0 | 0 |
| Dogan, K., Netherlands 2015 | Prospective | ERP | 75 | 48.4 (8.9) | 23 (30.7) | 44.9 (5.5) | 30 (40.0)[b] | 30 (40.0) | 18 (24.0) | 9 (12.0) | NR | NR | NR | NR | NR | NR | NR | NR |
| | | CC | 75 | 46.2 (10.1) | 23 (30.7) | 46.8 (5.6) | 31 (41.3)[b] | 35 (46.7) | 20 (26.7) | 13 (17.3) | NR | NR | NR | NR | NR | NR | NR | NR |
| Simonelli, V. Luxembourg, 2016 | Prospective | ERP | 90 | 42.1 (11.8)[c] | NR (25)[c] | 44.8 (5.9)[c] | NR | NR | NR | NR (34)[c] | 9 (8.7) | NR | NR | NR | NR | NR | NR | NR |
| | | CC | 90 | 41.5 (10.0)[c] | NR (28)[c] | 44.3 (5.8)[c] | NR | NR | NR | NR (26)[c] | 8 (7.7) | NR | NR | NR | NR | NR | NR | NR |
| Geubbels, N. Netherlands, 2019 | Randomised controlled | ERP | 110 | 42.7 (10.5) | 12 (10.9) | 42 (35.2–56.8)[a] | 18 (16.4)[b] | 39 (35.5) | 25 (22.7) | 6 (5.5) | NR | 16 (14.5) | 3 (2.7) | 2 (1.8) | 3 (2.7) | NR | NR | NR |
| | | CC | 110 | 42.6 (10.8) | 16 (14.5) | 41.4 (35–56)[a] | 16 (14.5)[b] | 35 (31.8) | 23 (20.9) | 9 (8.2) | NR | 16 (14.5) | 6 (5.5) | 1 (0.9) | 0 | NR | NR | NR |
| Mannaerts, G. United Arab Emirates, 2019 | Retrospective | ERP | 166 | 38.1 (12.1) | 43 (25.9) | 42.8 (5.6) | 59 (35.5) | 58 (34.9) | 57 (34.3) | 22 (13.3) | 4 (2.4) | 19 (11.4) | 51 (30.7) | 0 | 2 (1.2) | 1 (0.6) | 2 (1.2) | NR |
| | | CC | 48 | 36.1 (9.6) | 11 (22.9) | 46.5 (5.9) | 28 (58.3) | 18 (37.5) | 12 (25) | 17 (35.4) | 0 | 4 (8.3) | 3 (6.3) | 1 (2.1) | 0 | 1 (2.1) | 0 | NR |
| Ruiz-Tovar, J. Spain, 2019 | Randomised controlled | ERP | 90 | 45.3 (11.7) | 25 (27.8) | 44.9 (5.5) | NR (30) | NR (38.9) | NR (40) | NR (62.2) | NR | NR | NR | NR | NR | NR | NR | NR |
| | | CC | 90 | 44.8 (10.8) | 25 (27.8) | 44.5 (4.2) | NR (27.7) | NR (41.1) | NR (37.8) | NR (66.7) | NR | NR | NR | NR | NR | NR | NR | NR |

BMI: Body mass index; CC: Conventional care; ERP: Enhanced recovery programme; NR: Not reported; SD: Standard deviation.

[a]Median (range).

[b]Type 2 diabetes reported only.

[c]Characteristics are reported for the combined group of patients undergoing LRYGB (N = 90) and LSG (N = 13) procedures.

| Study | Care | Preoperative information, education and counselling | Prehabilitation and exercise | Smoking and alcohol cessation | Preoperative weight loss | Glucocorticoids | Preoperative fasting | Carbohydrate loading | Perioperative fluid management | Postoperative nausea and vomiting | Standardised anaesthetic protocol | Airway management | Ventilation strategies | Neuromuscular block | Monitoring of anaesthetic depth | Laparoscopy | Nasogastric tube | Abdominal drainage | Postoperative analgesia | Thromboprophylaxis | Early postoperative nutrition | Postoperative oxygenation | Non-invasive positive pressure ventilation |
|---|---|---|---|---|---|---|---|---|---|---|---|---|---|---|---|---|---|---|---|---|---|---|---|
| Geubbels, N. et al, 2014 | ERP | UTD | UTD | UTD | Y | UTD | UTD | UTD | UTD | UTD | Y | UTD | UTD | UTD | UTD | Y | Y | UTD | UTD | UTD | UTD | UTD | UTD |
|  | CC | UTD | UTD | UTD | Y | UTD | UTD | UTD | UTD | UTD | N | UTD | UTD | UTD | UTD | Y | UTD | UTD | UTD | UTD | UTD | UTD | UTD |
| Dogan, K. et al, 2015 | ERP | Y | UTD | UTD | UTD | N | UTD | UTD | UTD | Y | Y | Y | Y | UTD | UTD | Y | UTD | UTD | Y | Y | UTD | UTD | UTD |
|  | CC | Y | UTD | UTD | UTD | N | UTD | UTD | UTD | UTD | N | UTD | Y | UTD | UTD | Y | UTD | UTD | UTD | Y | UTD | UTD | UTD |
| Simonelli, V. et al, 2016 | ERP | Y | UTD | UTD | UTD | UTD | UTD | Y | Y | Y | Y | UTD | UTD | UTD | UTD | Y | Y | Y | Y | Y | UTD | UTD | UTD |
|  | CC | UTD | UTD | UTD | UTD | UTD | UTD | N | N | UTD | N | UTD | UTD | UTD | UTD | Y | N | Y | N | Y | UTD | UTD | UTD |
| Geubbels, N. et al, 2019 | ERP | Y | UTD | Y | Y | N | UTD | UTD | Y | Y | Y | UTD | UTD | UTD | UTD | Y | Y | UTD | Y | UTD | UTD | UTD | UTD |
|  | CC | N | UTD | Y | Y | UTD | UTD | UTD | N | UTD | Y | UTD | UTD | UTD | UTD | Y | N | UTD | N | UTD | UTD | UTD | UTD |
| Mannaerts, G. et al, 2019 | ERP | UTD | UTD | UTD | UTD | Y | UTD | UTD | UTD | Y | Y | UTD | UTD | UTD | UTD | Y | UTD | Y | Y | Y | UTD | UTD | UTD |
|  | CC | UTD | UTD | UTD | UTD | UTD | UTD | UTD | UTD | N | UTD | UTD | UTD | UTD | UTD | Y | UTD | N | UTD | UTD | UTD | UTD | UTD |
| Ruiz-Tovar, J. et al, 2019 | ERP | Y | UTD | UTD | Y | UTD | Y | UTD | Y | Y | Y | Y | UTD | UTD | UTD | Y | Y | Y | Y | Y | UTD | UTD | UTD |
|  | CC | UTD | UTD | UTD | N | UTD | N | UTD | N | Y | Y | Y | UTD | UTD | UTD | Y | N | N | N | Y | UTD | UTD | UTD |

**Fig 2. Summary of ERP implementation in included studies based on ERAS® Society recommendations.** Categories are based on elements from the Enhanced Recovery After Surgery (ERAS®) Society guidelines for bariatric surgery. CC: Conventional care; ERP: Enhanced recovery programme; N: Not implemented as per ERAS® recommendations; UTD: Unable to determine; Y: Implemented as per ERAS® recommendations.

## Reoperations

Reoperation rates were reported in two of the included studies (Table 1) [31,41]. The prospective observational study by Simonelli et al. (2016) reported comparable reoperation rates in the ERP group versus the conventional care group (10.0% [9/90] vs 7.7% [7/90]) within one month of follow-up, primarily for internal hernia [41]. The retrospective study by Mannaerts et al. (2019) did not report significant differences in 30-day reoperation rates for ERP versus conventional care (2.4% [4/166] vs 0.0% [0/48], p = 0.278) [31].

## Discussion

This review explored what evidence was available for the impact that ERPs have on outcomes in patients undergoing three common bariatric procedures. This follows the introduction of

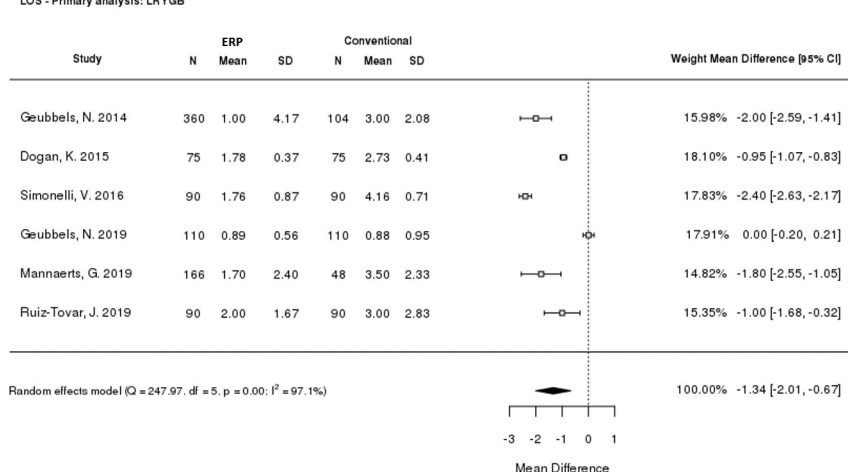

**Fig 3. Meta-analysis of length of stay.** CI: Confidence interval; ERP: Enhanced recovery programme; LOS: Length of stay; LRYGB: Laparoscopic Roux-en-Y gastric bypass; SD: Standard deviation.

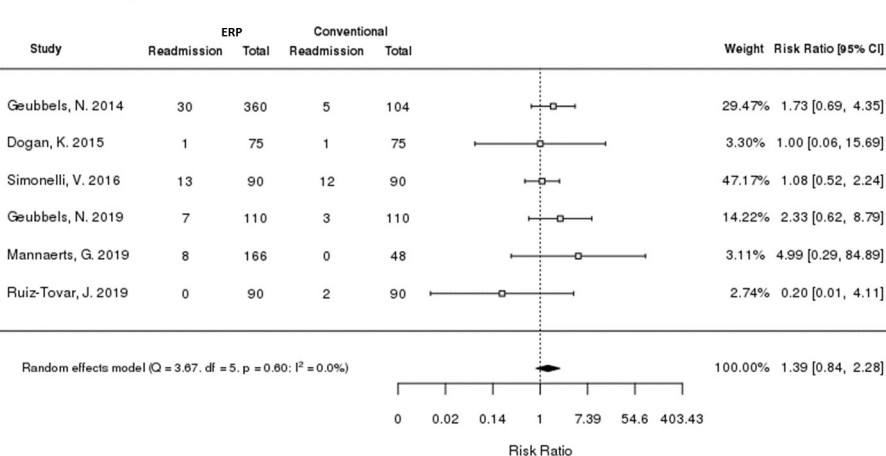

**Fig 4. Meta-analysis of 30-day readmission rate.** CI: Confidence interval; ERP: Enhanced Recovery Programme; LRYGB: Laparoscopic Roux-en-Y gastric bypass.

ERPs, with the increasing demand for bariatric surgery leading to a need for more efficient care. As only one relevant study was identified for each of LSG and OAGB, meta-analysis was not possible, and results are therefore not discussed for these procedures.

Meta-analysis of the six studies reporting data for LRYGB included in this SLR revealed that hospital length of stay was significantly shorter for patients receiving ERP care than for those receiving conventional care after LRYGB surgery. This result was confirmed with a sensitivity analysis excluding retrospective studies. A high level of heterogeneity was seen in the

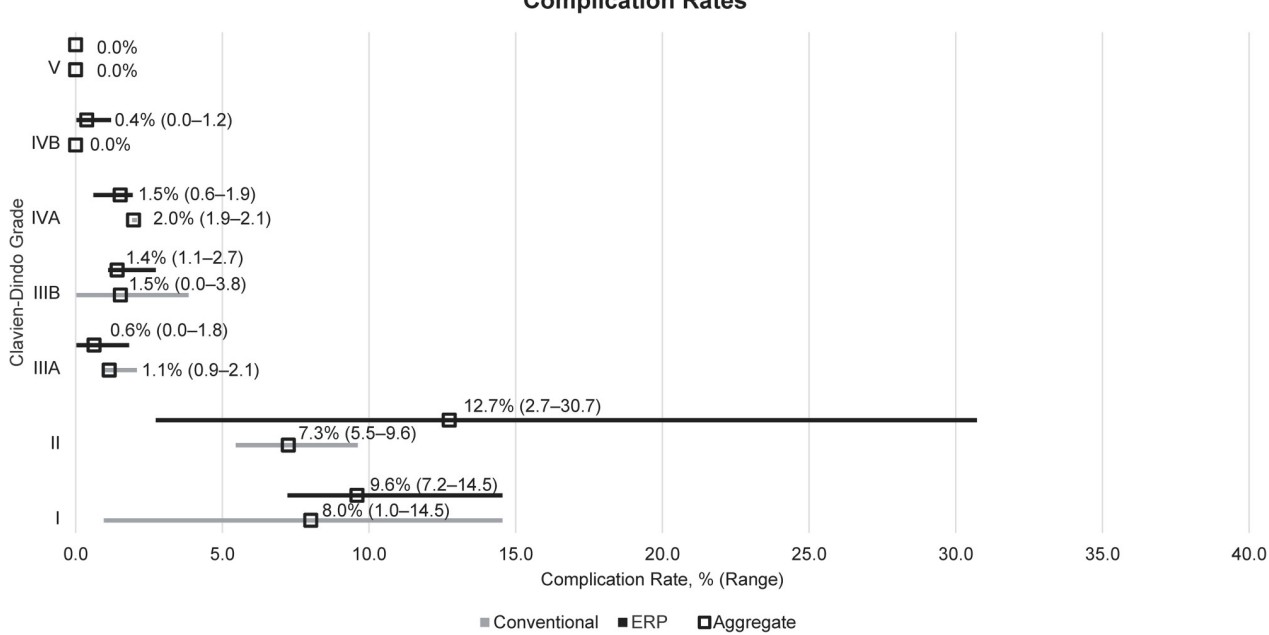

**Fig 5. Descriptive analysis of complication rates by Clavien-Dindo grade.** Aggregated rates were calculated by dividing the number of events reported in each study by the total number of patients across all studies. ERP: Enhanced recovery programme; LRYGB: Laparoscopic Roux-en-Y gastric bypass.

meta-analysis which is indicative of variation in length of stay across the included studies, as well as the different study types included (retrospective, prospective, and RCTs). Sources of variation could include differences in how conventional care and ERP groups were selected in different studies and availability of guidelines. Two studies evaluated concurrent ERP and conventional care groups [30,32]. The remaining four studies compared outcomes before and after an ERP was implemented [31,36,40,41]. As adoption of ERPs may vary, with some elements implemented before others and differing rates of adoption across hospitals and countries, what constitutes 'conventional care' in one study may more closely resemble an 'ERP' in another. This may explain why in the study by Geubbels et al. (2019) both treatment arms have a length of stay under 24 hours whereas all other studies report a length of stay of at least 1 day in either arm [30]. Geubbels et al. (2019) reported no significant difference in median total length of stay between ERP (21.3 hours) and conventional care (21.2 hours) groups, however the primary outcome was median functional hospital stay (time from the end of surgery until all discharge criteria had been met) which was significantly shorter with the ERP (17.4 vs 20.5 hours, p<0.001), suggesting that patients in the ERP group could have been discharged even earlier [30]. An additional factor could be variation in the discharge criteria used, although where reported these were broadly similar across the studies and therefore unlikely to be a major source of variation (S12 Table).

This review found no significant difference in 30-day readmission rates between ERPs and conventional care for LRYGB. Although meta-analysis reported a risk ratio of >1 for ERPs versus conventional care following LRYGB, this was not statistically significant, and moved closer to 1 upon sensitivity analysis. No individual study had a significantly different readmission rate for ERP versus conventional care, and very little heterogeneity was seen in the reporting of readmission rates, in contrast to the length of stay results.

Also analysed were 30-day complication rates after surgery. Only three studies reported complication rates by individual Clavien-Dindo grades, although another study did report 'specific' (Grade III+) and 'general' (Grade I–II) complications [41]. Lower-grade complications appeared to occur more frequently in patients receiving ERP care, however further evidence needs to be collected in this area before conclusions can be drawn over the relative safety of ERP and conventional care approaches.

The results of this review align with those of previous meta-analyses [28,29]. Although this meta-analysis included more up-to-date information, and analysed results from one of the most common bariatric procedures (namely LRYGB) rather than a combination of bariatric procedures, the overall results and conclusions were similar. The two previous meta-analyses, Małczak et al. (2017) and Ahmed et al. (2018), reported significantly shorter length of stay with ERPs compared with conventional care (mean difference 2.4 and 1.5 days, respectively; compared with 1.3 days in this meta-analysis), but also share significant heterogeneity in the reporting of length of stay results [28,29]. Our findings were also similar to Małczak et al. (2017), in that this previous meta-analysis did not report statistically significant differences in readmission rates between ERPs and conventional care, with a risk ratio of 0.86 (95% CI: 0.57, 1.30) for ERP versus conventional care [28].

Despite the addition of more recent studies comparing ERPs and conventional care methods, new conclusions could not be drawn, and the evidence gaps seen in previous reviews remains. Few relevant studies were identified in the review and those included were of mixed study designs, including only two RCTs. A high level of heterogeneity was seen between studies, particularly when comparing duration of hospital-stay. Surprisingly, considering the relative frequency with which sleeve gastrectomy and OAGB procedures are performed in EMEA, only individual studies reporting relevant comparative outcomes with LSG and OAGB were identified, representing a major evidence gap [25].

This SLR was a broad, reproducible search of the literature to identify recent clinical and economic evidence related to ERPs in LRYGB, OAGB and LSG within EMEA. Searches were designed to capture a wide range of study designs, however, only studies involving an ERP compared to conventional care were included. As there is variation in the specific protocols employed by different hospitals and practitioners, as well as in the quality of reporting how ERPs were implemented, this could have impacted the SLR results, particularly as evidence was identified from only four European and Middle Eastern countries. Indeed the extent to which elements from the ERAS® Society guidelines were implemented was not extensively reported in the studies identified by this review, and even with the use of a date limit to try to capture current clinical practice, there remained some differences between older and more recent studies. The ERAS® Society published the RECOvER Checklist in 2019 in an effort to standardise the reporting of ERPs in the literature which may help to address some of this variation in the future [43]. Finally, while articles written in any European language were eligible for inclusion and no articles were ultimately excluded from the review as a result of the publication language, the search strategy for this SLR was in the English language and databases with a predominant English-language focus were used, which may have contributed to the lack of publications identified from countries where English is not a predominant language.

## Conclusions

The results of this review support the use of ERPs in terms of faster time to discharge, allowing greater turnover of patients, especially in locations where hospital space is limited, whereas analyses of readmission, reoperation and complication rates were inconclusive. This review has demonstrated that there remains a scarcity of robust evidence on the impact of bariatric ERPs, both in terms of the lack of standardised assessment of outcomes in studies as well as inconsistent reporting, and potentially implementation, of ERPs. Further research is needed with larger numbers of patients undergoing each procedure type and with better standardisation of the assessment of complications and clinical outcomes in order to confirm the findings for shorter length of stay, to conclude whether ERPs have a different associated risk when compared to conventional care, and to fill in the data gaps relating to ERPs for LSG and OAGB specifically.

## Supporting information

**S1 Checklist.**
(DOCX)

**S1 Fig. Length of stay meta-analysis sensitivity analysis (excluding studies with estimated mean and standard deviation values).** CI: Confidence interval; ERP: Enhanced recovery programme; LOS: Length of stay; RCT: Randomised controlled trial; SD: Standard deviation.
(TIF)

**S2 Fig. Length of stay meta-analysis sensitivity analysis (excluding retrospective studies).** CI: Confidence interval; ERP: Enhanced recovery programme; LOS: Length of stay; RCT: Randomised controlled trial; SD: Standard deviation.
(TIF)

**S3 Fig. 30-day readmission meta-analysis sensitivity analysis (excluding retrospective studies).** CI: Confidence interval; ERP: Enhanced recovery programme; RCT: Randomised controlled trial.
(TIF)

**S1 Table. PRISMA checklist.**
(DOCX)

**S2 Table. Search terms for MEDLINE, MEDLINE in-process, MEDLINE Epub ahead of print and Embase–original review.** Orthopaedic search terms are included due to the intended original scope of this systematic literature review including orthopaedic surgery.
(DOCX)

**S3 Table. Search terms for MEDLINE, MEDLINE in-process, MEDLINE Epub ahead of print and Embase–update review.**
(DOCX)

**S4 Table. Search terms for the Cochrane library databases (searched via the Wiley Online Platform)–original review.** Orthopaedic search terms are included due to the intended original scope of this systematic literature review including orthopaedic surgery.
(DOCX)

**S5 Table. Search terms for the Cochrane library databases (searched via the Wiley Online Platform)–update review.**
(DOCX)

**S6 Table. Search terms for grey literature sources.** ERAS: Enhanced Recovery After Surgery; ERP: Enhanced recovery programme. Orthopaedic search terms are included due to the intended original scope of this systematic literature review including orthopaedic surgery.
(DOCX)

**S7 Table. Eligibility criteria for the SLR.** BMI: Body mass index; EMEA: Europe, the Middle East and Africa; EQ-5D: EuroQoL– 5 Dimensions; ERP: Enhanced recovery programme; FT: Fast track; RCT: Randomised controlled trial; SF-36: 36-Item Short Form Survey; SLR: Systematic literature review.
(DOCX)

**S8 Table. Information captured in the extraction grid.** ASA: American Society of Anesthesiologists; BMI: Body mass index; EQ-5D: EuroQoL– 5 Dimensions; SF-36: 36-Item Short Form Survey.
(DOCX)

**S9 Table. Modified downs and black checklist used for risk of bias assessment of non-randomised studies.**
(DOCX)

**S10 Table. OAGB and LSG Data excluded from the analyses.** BMI: Body mass index; CC: Conventional care; ERP: Enhanced recovery programme; NR: Not reported; SD: Standard deviation. [a]Type 2 diabetes reported only.
(DOCX)

**S11 Table. Risk of bias assessment results.**
(DOCX)

**S12 Table. Discharge criteria reported in included studies.** CRP: C-reactive protein; VAS: Visual analogue scale.
(DOCX)

## Acknowledgments

The authors acknowledge Rawan Abu Hasan and Saqer Al-Deraan for their contributions to the study conception and design, analysis and interpretation of data and support in the content and preparation of this manuscript.

The authors acknowledge Shahad Atrah (Costello Medical, London, UK); David Pritchett, Lirazel Swindells and Sarah Wayman (Costello Medical, Cambridge, UK) for their support in performing the literature review, and Daniel Smith and Amelia Frizell-Armitage (Costello Medical, Cambridge, UK) for medical writing support and editorial assistance in preparing this manuscript for publication, based on authors' input and direction

## Author Contributions

**Conceptualization:** Cindy Tong, Hannah Taylor, Karl Miller, Thao Nguyen Phan Thanh, Christian Ridley, Sara Steeves, William Marsh.

**Data curation:** Khalid Al-Rubeaan, Cindy Tong, Hannah Taylor, Christian Ridley, Sara Steeves, William Marsh.

**Formal analysis:** Khalid Al-Rubeaan, Cindy Tong, Hannah Taylor, Sara Steeves.

**Funding acquisition:** Hannah Taylor, Thao Nguyen Phan Thanh.

**Investigation:** Khalid Al-Rubeaan, Cindy Tong, Hannah Taylor, Karl Miller, Thao Nguyen Phan Thanh, Christian Ridley, Sara Steeves, William Marsh.

**Methodology:** Khalid Al-Rubeaan, Cindy Tong, Hannah Taylor, Karl Miller, Thao Nguyen Phan Thanh, Christian Ridley, Sara Steeves, William Marsh.

**Project administration:** Hannah Taylor, Christian Ridley, Sara Steeves, William Marsh.

**Resources:** Hannah Taylor.

**Supervision:** Hannah Taylor, Thao Nguyen Phan Thanh.

**Visualization:** Cindy Tong, Sara Steeves.

**Writing – original draft:** Khalid Al-Rubeaan, Cindy Tong, Hannah Taylor, Sara Steeves.

**Writing – review & editing:** Khalid Al-Rubeaan, Cindy Tong, Hannah Taylor, Karl Miller, Thao Nguyen Phan Thanh, Christian Ridley, Sara Steeves, William Marsh.

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
