## [Decision Letter · Decision Letter 0]

10 Aug 2020

PONE-D-20-17131

Enhanced Recovery Programmes Versus Conventional Care in Bariatric Surgery: A Systematic Literature Review and Meta-Analysis

PLOS ONE

Dear Dr. Steeves,

Thank you for submitting your manuscript to PLOS ONE. After careful consideration, we feel that it has merit but does not fully meet PLOS ONE’s publication criteria as it currently stands. Therefore, we invite you to submit a revised version of the manuscript that addresses the points raised during the review process.

We look forward to receiving your revised manuscript.

Kind regards,

Ahmed Negida, MD

Academic Editor

PLOS ONE

Journal Requirements:

"I have read the journal's policy and the authors of this manuscript have the following competing interests:

CT, HT, KM and TN: Employees of Johnson & Johnson

KA: Employee of Sultan Bin Abdulaziz Humanitarian City at the time this study was conducted, and paid consultant for Johnson & Johnson

CR, SS and WM: Employees of Costello Medical at the time this study was conducted and have served as paid consultants for Johnson & Johnson"

We note that one or more of the authors have an affiliation to the commercial funders of this research study : Johnson & Johnson Medical NV.

We also note that one or more of the authors are employed by a commercial company: Costello Medical Consulting Limited.

2.1. Please provide an amended Funding Statement declaring these commercial affiliations, as well as a statement regarding the Role of Funders in your study. If the funding organization did not play a role in the study design, data collection and analysis, decision to publish, or preparation of the manuscript and only provided financial support in the form of authors' salaries and/or research materials, please review your statements relating to the author contributions, and ensure you have specifically and accurately indicated the role(s) that these authors had in your study. You can update author roles in the Author Contributions section of the online submission form.

2.2. Please also provide an updated Competing Interests Statement declaring this commercial affiliation along with any other relevant declarations relating to employment, consultancy, patents, products in development, or marketed products, etc. 

3. Please include captions for figure  4 and 5.

5. We note that this manuscript is a systematic review or meta-analysis; our author guidelines therefore require that you use PRISMA guidance to help improve reporting quality of this type of study. Please upload copies of the completed PRISMA checklist as Supporting Information with a file name “PRISMA checklist”.

Reviewers' comments:

Reviewer's Responses to Questions

**Comments to the Author**

1. Is the manuscript technically sound, and do the data support the conclusions?

Reviewer #1: Yes

Reviewer #2: Yes

Reviewer #3: Yes

Reviewer #4: Yes

2. Has the statistical analysis been performed appropriately and rigorously? 

Reviewer #1: Yes

Reviewer #2: Yes

Reviewer #3: Yes

Reviewer #4: I Don't Know

3. Have the authors made all data underlying the findings in their manuscript fully available?

Reviewer #1: Yes

Reviewer #2: Yes

Reviewer #3: Yes

Reviewer #4: No

4. Is the manuscript presented in an intelligible fashion and written in standard English?

Reviewer #1: Yes

Reviewer #2: Yes

Reviewer #3: No

Reviewer #4: Yes

5. Review Comments to the Author

Reviewer #1: This meta-analysis aims to assess the utility of enhanced recovery programs (ERAS) following bariatric surgery. Bariatric surgery is indicated in certain cases of type 2 diabetes and metabolic syndrome, and enhanced recovery programs may play a critical role in improving the post-operative experience of the patient. Given the degree of importance & prevalence of the aforementioned conditions, the authors’ choice of topic is strongly commended. There are already two recent meta analyses covering this topic by Malczak and Ahmed et al. published in 2017 and 2018 respectively. The present review corroborates the findings of the previous two meta-analyses, though it does not add a significant novel finding of its own, with all 3 reviews finding a significant reduction in hospitalization duration but non-significant differences with respect to complications/readmission rate. The authors state that the novel aspect of this review is that it aims to stratify by type of bariatric procedure; unfortunately, the only procedure for which multiple (six) studies were available was Roux-en-Y bypass, and was thus the only procedure to undergo meta-analysis. The other two procedures (sleeve gastrectomy and one anastomosis gastric bypass) each had one study and were not further discussed in the manuscript as per the authors. Given the existing limitation within the literature, the primary promise of the meta-analysis (investigating the effect of recovery programs stratified by type of procedure) could not be achieved.

I believe the quality of the methods section, and therefore of the review as a whole, would improve by addressing the following points:

• The authors referenced the paper by Hozo et al. in estimating the SD from the range (Range / 6); however, they did not clarify the sample sizes to which this formula would be applied, as it was recommended by Hozo et al. in cases where N > 70. I believe the manuscript would benefit from an increased degree of clarity if the authors were to clarify this.

• The authors state that they used the median as an estimate for mean values in some studies. It would be ideal if the authors were to specify exactly which studies used the median as well as conduct a sensitivity analysis excluding said studies. This is because the median may have been used by the authors of the original studies because of a skewed -rather than normal- distribution of data; therefore, simply replacing it by the mean may not be appropriately representative of the study’s results. This is especially likely to be an issue if included studies had a small sample size. Conducting a sensitivity analysis will likely improve the robustness of the findings and improve the overall quality of the manuscript.

• One concern is the focus on studies conducted in Europe, the Middle East and Africa while excluding non-English studies. Due to the linguistic variation in the aforementioned regions, this may well be a significant limitation in the findings of the review (though this limitation is acknowledged by the reviewers). Had the language restriction been removed, this may have aided the authors in finding a sufficient number of studies to perform a meta-analysis for the other two procedures (though this is difficult to ascertain since the PRISMA diagram does not specify the number of studies excluded due to the English-only limitation). If it is possible to retrieve such studies, I believe this would enhance the credibility of the review.

• Though standardized checklists are often modified to fit the purposes of each individual review, it would probably be beneficial if the authors were to clarify what modifications were done to the Downs and Blacks checklist, as this may impact the quality assessment of the included studies and quell any concerns other reviewers may have.

• The study is otherwise well-written and structured, and the PRISMA diagram is especially well-done.

Two previous meta-analyses:

1. Ahmed OS, Rogers AC, Bolger JC, Mastrosimone A, Robb WB. Meta-analysis

of enhanced recovery protocols in bariatric surgery. J Gastrointest Surg.

2018;22(6):964-972.

2. Małczak P, Pisarska M, Piotr M, Wysocki M, Budzynski A, Pedziwiatr M.

Enhanced Recovery after Bariatric Surgery: Systematic Review and Meta-Analysis.

Obes Surg. 2017;27(1):226-235. Epub 2016/11/07. doi: 10.1007/s11695-016-2438-

z. PubMed PMID: 27817086; PubMed Central PMCID: PMCPMC5187372.

Reviewer #2: This is a systematic review and meta-analysis in which the authors compare the efficacy of Enhanced Recovery Programmes Versus Conventional Care in Bariatric Surgery. This is considered an updated review which supports the enhanced recovery programmes which allow shorter hospital stay. Although the study is inconclusive, it enhances the future research to fill the current data gaps. The manuscript is well-written and structure and the quality of research is scientifically sound.

Reviewer #3: The authors performed a systematic review with meta-analysis to evaluate the effects of enhanced recovery programmes after three common bariatric procedures: laparoscopic Roux en-Y gastric bypass, laparoscopic sleeve gastrectomy, and one anastomosis gastric bypass.

Although the overall approach to the review is proper, I have few comments:

1. Abstract

- Replace (Method) with (Methods). Also, replace (papers) with (studies).

- Insert the software used for analysis.

- Insert the total number of included patients.

- P-values should be reported.

2. Introduction

- Line 65: insert citation for the previous meta-analyses, and highlight the unique data reported in this systematic review as well as any controversies.

3. Methods

- The databases were searched up to 2019; please consider updating it.

- Insert the software used for analysis.

4. Results

- line 141: insert reference of the included studies.

- Line 174-176: mention that the sensitivity analysis did not resolve the heterogeneity. Was the subgroup analysis performed?

5. Language: The entire manuscript needs professional revision for grammatical errors and stylistic editing. For example,

- Line 2: (it is likely the demand), Squinting modifier!!

- Line 4: (XX and more efficient use) & (XX and one anastomosis gastric), insert a comma before (and).

- Line 15: (outcomes, however as) should be (outcomes; however, as).

- (Meta-analysis revealed), consider adding an article (The).

- Line 48, 51, etc.: The comma before the reference number should be replaced with a dot.

Reviewer #4: Review of the manuscript “Enhanced Recovery Programmes Versus Conventional Care in Bariatric Surgery: A Systematic Literature Review and Meta-Analysis”

Since the end of the 90s of the last century, there have been many reports describing a method defined as ERPs. The strategy of fast track gathers various elements of perioperative procedure. It takes into account the pathophysiology of operation injury and eliminates surgery procedures that are not justified in the perspective of evidence-based medicine. Optimal preparation of a patient for the operation connected with oral and written information about the surgical procedure and postoperative course, early feeding and rehabilitation on the day of surgery and optimal pain control make up the most important elements of pre- and post-operative procedures based on fast track surgery. The intraoperative factors include minimal-access surgery, thoracic epidural anesthesia and no routine use of nasogastric tube and abdominal drains. The effect of such a procedure is a decrease of postoperative complications, improvement of patient’s comfort and satisfaction, and shortening of hospital stay, and at the same time, reduction of hospitalization costs.

The authors presented systematic literature review aimed to evaluate the effects of enhanced

recovery programmes after LRYGB. The results can be drawn that using ERP in the obese patients qualified to LRYGB shortens length of hospital stay.

The study design is clear and well described. The used methods are generally acceptable.

The topic of article is interesting, methods of the study are presented concisely.

My remarks:

1. In figure 2 authors characterised of ERP implementation for included studies. Analysis of fig 2 suggest that data about important elements of ER programme are unable to determine in most analysed articles. It is difficult to compare the results precisely since the trials applied variable protocols of enhanced recovery programs and each study assumed various final points. In my opinion it should be cleary described and highlights in the manuscript.

2. Although we have an access to a few study on ERPs, it is difficult to prove, in an unambiguous manner, an advantage of one of the methods. It seems that relaying on the presented results, a conclusion can be drawn that using ERPs shortens length of hospital stay, but further RCTs study are necessary.

Authors concluded that “…results of this review support the use of ERPs in terms of faster time to discharge…”.

What was criteria of discharging a patient from hospital in analyzed study? There are same or differ in these study, it can impacts of obtained results.

3.References should be checked according to submission guidelines

6. PLOS authors have the option to publish the peer review history of their article (what does this mean?). If published, this will include your full peer review and any attached files.

Reviewer #1: No

Reviewer #2: No

Reviewer #3: No

Reviewer #4: No

---

## [Author Response · Author response to Decision Letter 0]

21 Sep 2020

We thank the reviewers for their helpful feedback on the manuscript. Below are the changes we have made to address their comments. Please note that line numbers refer to the ‘Track Changes’ version of the manuscript.

Reviewer #1:

COMMENT - The authors referenced the paper by Hozo et al. in estimating the SD from the range (Range / 6); however, they did not clarify the sample sizes to which this formula would be applied, as it was recommended by Hozo et al. in cases where N > 70. I believe the manuscript would benefit from an increased degree of clarity if the authors were to clarify this. RESPONSE - Thank you, we have clarified that the N>70 version was applicable to the studies included in the analysis. [L137-139]

COMMENT - The authors state that they used the median as an estimate for mean values in some studies. It would be ideal if the authors were to specify exactly which studies used the median as well as conduct a sensitivity analysis excluding said studies. This is because the median may have been used by the authors of the original studies because of a skewed -rather than normal- distribution of data; therefore, simply replacing it by the mean may not be appropriately representative of the study’s results. This is especially likely to be an issue if included studies had a small sample size. Conducting a sensitivity analysis will likely improve the robustness of the findings and improve the overall quality of the manuscript. RESPONSE - Thank you for these suggestions. We have added citations for the studies where the mean and standard deviation needed to be estimated to the Methods section. A sensitivity analysis excluding these studies has also been performed and is included as Supplementary Figure 1; the results are the same as the primary analysis including all studies. [L140-141]

COMMENT - One concern is the focus on studies conducted in Europe, the Middle East and Africa while excluding non-English studies. Due to the linguistic variation in the aforementioned regions, this may well be a significant limitation in the findings of the review (though this limitation is acknowledged by the reviewers). Had the language restriction been removed, this may have aided the authors in finding a sufficient number of studies to perform a meta-analysis for the other two procedures (though this is difficult to ascertain since the PRISMA diagram does not specify the number of studies excluded due to the English-only limitation). If it is possible to retrieve such studies, I believe this would enhance the credibility of the review. RESPONSE - Thank you for raising this. The review included publications in any European language rather than using an English language limit, which we have clarified in the Methods section. No articles were ultimately excluded as a result of the language limit however, which we have now mentioned in the Discussion section. [L109-114; L303-305]

COMMENT - Though standardized checklists are often modified to fit the purposes of each individual review, it would probably be beneficial if the authors were to clarify what modifications were done to the Downs and Blacks checklist, as this may impact the quality assessment of the included studies and quell any concerns other reviewers may have. RESPONSE - Thank you for this suggestion. We have added more information about how the checklist was modified to the Methods section and included the modified version of the checklist as Supplementary Table 9. [L126-128]

COMMENT - The study is otherwise well-written and structured, and the PRISMA diagram is especially well-done. RESPONSE - We thank the reviewer for their feedback.

Reviewer #2:

COMMENT - This is a systematic review and meta-analysis in which the authors compare the efficacy of Enhanced Recovery Programmes Versus Conventional Care in Bariatric Surgery. This is considered an updated review which supports the enhanced recovery programmes which allow shorter hospital stay. Although the study is inconclusive, it enhances the future research to fill the current data gaps. The manuscript is well-written and structure and the quality of research is scientifically sound. RESPONSE - We thank the reviewer for their feedback.

Reviewer #3:

COMMENT - Replace (Method) with (Methods). Also, replace (papers) with (studies). RESPONSE - Thank you, these changes have been made [L9-10]

COMMENT - Insert the software used for analysis. RESPONSE - We have added that analyses were conducted in R [L14]

COMMENT - Insert the total number of included patients. RESPONSE - We have added this information [L15-17]

COMMENT - P-values should be reported. RESPONSE - We have added this information [L21-24]

COMMENT - Line 65: insert citation for the previous meta-analyses, and highlight the unique data reported in this systematic review as well as any controversies. RESPONSE - Thank you for this suggestion, we have added the citations to this sentence and provided the rationale for conducting a new review in the context of these previous meta-analyses. [L68-69]

COMMENT - The databases were searched up to 2019; please consider updating it. RESPONSE - Thank you for this suggestion. We have conducted targeted searches over the past year and do not believe that any major new data that would change the review’s conclusions have been published since the searches were last updated.

COMMENT - Insert the software used for analysis. RESPONSE - We have added that analyses were conducted in R [L136]

COMMENT - line 141: insert reference of the included studies. RESPONSE - We have added the citations

COMMENT - Line 174-176: mention that the sensitivity analysis did not resolve the heterogeneity. Was the subgroup analysis performed? RESPONSE - We have added that the high level of heterogeneity remained in the sensitivity analysis. No subgroup analyses beyond the sensitivity analyses described in the Methods section were performed, and all results of the sensitivity analyses are reported in full in the Supplementary material (Figures S1-S3). [L188]

COMMENT - Line 2: (it is likely the demand), Squinting modifier!! RESPONSE - We have removed this from the abstract [L2]

COMMENT - Line 4: (XX and more efficient use) & (XX and one anastomosis gastric), insert a comma before (and). RESPONSE - We have added commas in these places [L4-7]

COMMENT - Line 15: (outcomes, however as) should be (outcomes; however, as). RESPONSE - We have made this change [L17]

COMMENT - (Meta-analysis revealed), consider adding an article (The). RESPONSE - We have made this change [L19]

COMMENT - Line 48, 51, etc.: The comma before the reference number should be replaced with a dot. RESPONSE - We have moved all references in the manuscript to the end of the sentences they are in.

Reviewer #4:

COMMENT - In figure 2 authors characterised of ERP implementation for included studies. Analysis of fig 2 suggest that data about important elements of ER programme are unable to determine in most analysed articles. It is difficult to compare the results precisely since the trials applied variable protocols of enhanced recovery programs and each study assumed various final points. In my opinion it should be cleary described and highlights in the manuscript. RESPONSE - We have added further details to the Results section discussing Figure 2 to highlight the difficulty of making comparisons between the studies due to the lack of clear reporting. [L170-172]

COMMENT - Although we have an access to a few study on ERPs, it is difficult to prove, in an unambiguous manner, an advantage of one of the methods. It seems that relaying on the presented results, a conclusion can be drawn that using ERPs shortens length of hospital stay, but further RCTs study are necessary. RESPONSE - We have updated the Conclusions section to reflect the fact that there remains some uncertainty in the length of stay results, and that it would be beneficial to have additional studies to support the conclusion. [L312-320]

COMMENT - Authors concluded that “…results of this review support the use of ERPs in terms of faster time to discharge…”. RESPONSE - 

COMMENT - What was criteria of discharging a patient from hospital in analyzed study? There are same or differ in these study, it can impacts of obtained results. RESPONSE - Although not reported in one study, the discharge criteria used in the other studies were broadly similar. This was aligned with our expectations as the focus of the analyses was on a specific surgical procedure which should have generally universal discharge criteria. We have added a statement to the discussion section mentioning this as a possible source of variation and provided the criteria that were reported in each study in the supplementary material (Table S11). [L253-256]

COMMENT - References should be checked according to submission guidelines. RESPONSE - All of the references have been checked against the submission guidance

---

## [Decision Letter · Decision Letter 1]

19 Oct 2020

PONE-D-20-17131R1

Enhanced Recovery Programmes Versus Conventional Care in Bariatric Surgery: A Systematic Literature Review and Meta-Analysis

PLOS ONE

Dear Dr. Steeves,

Thank you for submitting your manuscript to PLOS ONE. After careful consideration, we feel that it has merit but does not fully meet PLOS ONE’s publication criteria as it currently stands. Therefore, we invite you to submit a revised version of the manuscript that addresses the points raised during the review process.

We look forward to receiving your revised manuscript.

Kind regards,

Ahmed Negida, MD

Academic Editor

PLOS ONE

Reviewers' comments:

Reviewer's Responses to Questions

**Comments to the Author**

1. If the authors have adequately addressed your comments raised in a previous round of review and you feel that this manuscript is now acceptable for publication, you may indicate that here to bypass the “Comments to the Author” section, enter your conflict of interest statement in the “Confidential to Editor” section, and submit your "Accept" recommendation.

Reviewer #1: (No Response)

Reviewer #3: All comments have been addressed

Reviewer #4: All comments have been addressed

2. Is the manuscript technically sound, and do the data support the conclusions?

Reviewer #1: Partly

Reviewer #3: Yes

Reviewer #4: Yes

3. Has the statistical analysis been performed appropriately and rigorously? 

Reviewer #1: No

Reviewer #3: Yes

Reviewer #4: Yes

4. Have the authors made all data underlying the findings in their manuscript fully available?

Reviewer #1: Yes

Reviewer #3: Yes

Reviewer #4: Yes

5. Is the manuscript presented in an intelligible fashion and written in standard English?

Reviewer #1: No

Reviewer #3: Yes

Reviewer #4: Yes

6. Review Comments to the Author

Reviewer #1: Abstract

1- Remove the number of words

2- The study objectives is different from the final conclusion; the authors aimed to evaluate the effects of enhanced recovery programmes on three types of bariatric surgeries; however, in the final conclusion they mentioned only the laparoscopic Roux-en-Y gastric bypass surgery. Even if you couldn't include other surgeries in the analysis, you have to highlight the current evidence regarding these surgeries in the conclusion, or you can adjust your objectives.

3- Line 16: add "comma" after "however".

4- Line 18: add "hospital" to be "duration of hospital-stay".

5- Add up to 6 keywords related to your study.

Introduction

6- Line 35: Cite each mentioned guidelines.

7- The authors highlighted the association between obesity and T2DM, I suggest adding a hint about the overall morbidity and mortality associated with obesity before talking about T2DM.

8- Line 47: I suggest adding one sentence about the impact of bariatric surgery on insulin resistance here.

9- The authors need to clearly present their rationale to conduct this study, as the previous two meta-analysis has similar objectives with larger sample and similar findings.

Methods

10- Please cite the protocol of this study, if you registered it in PROSPERO, or published it on an online repository.

11- The last update of the study search was 15 months ago, I strongly recommend updating this search and preferably include Scopus database in your search to ensure including all potential studies.

12- Did you exclude the seventh study that includes data regarding the other surgeries?

If yes (I think so), you have to adjust the objectives to include only reported surgery (as I mentioned before). If no, you have to report it's data and include it in the tables of summary. Moreover, the PRISMA flow diagram should be adjusted to include only 6 studies in the final stage. The number of included studies in the PRISMA should match the exact number of studies in the summary tables and Quality assessment.

13- The PRISMA flow diagram should be adjusted to include the reasons of exclusion of the last 11 and 4 studies.

14- Do you have any explanation for using RR instead of OR in readmission?

15- Consider to conduct a subgroup analysis in the LOS outcome according to the study design to solve the heterogeneity.

Discussion

16- "This review explored the evidence available for the impact that ERPs have on outcomes in patients undergoing three common bariatric procedures". You evaluated the impact of ERPs on only one surgery not three, please correct.

17- "As only one relevant study was identified for each of LSG and OAGB, meta-analysis was not possible, and results are therefore not

discussed for these procedures." State clearly that you exclude the study from your review, as you did not report and data regarding it.

18- Line 258 "reporting recent clinical and economic evidence related to ERPs in LRYGB, OAGB and LSG within EMEA" You cannot consider this as a strength point as you did not report any data regarding OAGB and LSG.

19- Significant heterogeneity and the

relatively small number of studies should be added to the section of limitations.

20- The manuscript needs language editing to eliminate any mistakes.

Reviewer #3: The authors have addressed all my comments/suggestions.

Reviewer #4: I have no comments. I accepted author response to manuscripte Enhanced Recovery Programmes Versus Conventional Care in Bariatric Surgery: A Systematic Literature Review and Meta-Analysis

7. PLOS authors have the option to publish the peer review history of their article (what does this mean?). If published, this will include your full peer review and any attached files.

Reviewer #1: No

Reviewer #3: No

Reviewer #4: **Yes: **Maciej Wiewiora

---

## [Author Response · Author response to Decision Letter 1]

3 Nov 2020

We thank the reviewers for their helpful feedback on the manuscript. Below are the changes we have made to address their comments. Please note that line numbers refer to the ‘Track Changes’ version of the manuscript.

Reviewer #1:

COMMENT - [Abstract] 1- Remove the number of words. RESPONSE - Thank you, this change has been made [L1]

COMMENT - [Abstract] 2- The study objectives is different from the final conclusion; the authors aimed to evaluate the effects of enhanced recovery programmes on three types of bariatric surgeries; however, in the final conclusion they mentioned only the laparoscopic Roux-en-Y gastric bypass surgery. Even if you couldn't include other surgeries in the analysis, you have to highlight the current evidence regarding these surgeries in the conclusion, or you can adjust your objectives. RESPONSE - Thank you for this comment. We have adjusted the wording of the objectives to clarify the objective was to evaluate the available evidence on the effects of enhanced recovery programmes in all three procedures, and then updated the conclusions to state that the data gaps identified by the review include the lack of data for OAGB and LSG. [L5; L32]

COMMENT [Abstract] 3- Line 16: add "comma" after "however". RESPONSE - Thank you, this change has been made [L16]

COMMENT [Abstract] 4- Line 18: add "hospital" to be "duration of hospital-stay". RESPONSE - Thank you, this change has been made [L19]

COMMENT [Abstract] 5- Add up to 6 keywords related to your study. RESPONSE - These have been added after the abstract [L33-35]

COMMENT [Introduction] 6- Line 35: Cite each mentioned guidelines. RESPONSE - Thank you, we have added the missing citations [L41-42]

COMMENT [Introduction] 7- The authors highlighted the association between obesity and T2DM, I suggest adding a hint about the overall morbidity and mortality associated with obesity before talking about T2DM. RESPONSE - Thank you, we have added the following sentences before the association with T2DM is introduced: “Severe obesity can increase the risk of hypertension, hyperlipidaemia, heart disease and ischaemic stroke, as well as a number of cancers including cancer of the colon, gall bladder, rectum and liver. Obesity is also closely linked to the development of T2DM due to a progressive decrease in insulin secretion alongside a rise in insulin resistance.” [L44-48]

COMMENT [Introduction] 8- Line 47: I suggest adding one sentence about the impact of bariatric surgery on insulin resistance here. RESPONSE - Thank you for this suggestion, we have added a sentence that bariatric surgery has been shown to improve insulin resistance, referencing Schauer et al. 2012 [L55-56]

COMMENT [Introduction] 9- The authors need to clearly present their rationale to conduct this study, as the previous two meta-analysis has similar objectives with larger sample and similar findings. RESPONSE - We have added that the literature reviews to inform the previous meta-analyses were conducted several years ago (May 2017 and April 2016) to clarify that the rationale for our study was to review whether the evidence limitations noted in these original analyses are still present despite additional studies having since been published. [L77-78] 

COMMENT [Methods] 10- Please cite the protocol of this study, if you registered it in PROSPERO, or published it on an online repository. RESPONSE - Thank you for this suggestion. Our protocol was not registered in an online repository therefore we are unable to cite it. 

COMMENT [Methods] 11- The last update of the study search was 15 months ago, I strongly recommend updating this search and preferably include Scopus database in your search to ensure including all potential studies. RESPONSE - Thank you for this suggestion. We have conducted targeted searches over the past year and do not believe that any major new data that would change the review’s conclusions have been published since the searches were last updated. While we did not search the Scopus database, we think it would be unlikely to yield additional relevant studies in this case as we searched multiple electronic databases (including CENTRAL, MEDLINE and Embase as recommended by the Cochrane Handbook for Systematic Reviews of Interventions) and performed hand searches of citations in included studies, which is the suggested use of Scopus by the Cochrane Handbook (Section 4). 

COMMENT [Methods] 12- Did you exclude the seventh study that includes data regarding the other surgeries? If yes (I think so), you have to adjust the objectives to include only reported surgery (as I mentioned before). If no, you have to report it's data and include it in the tables of summary. Moreover, the PRISMA flow diagram should be adjusted to include only 6 studies in the final stage. The number of included studies in the PRISMA should match the exact number of studies in the summary tables and Quality assessment. RESPONSE - Yes, we excluded the seventh study from the analyses as it was the only study to report data for OAGB, however it was included in the review itself due to fulfilling the eligibility criteria. We have updated the PRISMA flow diagram to make this distinction clearer and now refer only to the studies that were ultimately included in the analyses as the final level of the diagram. [Fig 1] As the seventh study was not included in the analyses, we have added the characteristics and outcomes extracted from this study together with the data from the study reporting outcomes following LSG as a separate table in the data supplement (S10 Table). 

COMMENT [Methods] 13- The PRISMA flow diagram should be adjusted to include the reasons of exclusion of the last 11 and 4 studies. RESPONSE - Thank you for this suggestion, we have updated the diagram to include these reasons. [Fig 1]

COMMENT [Methods] 14- Do you have any explanation for using RR instead of OR in readmission? RESPONSE - We used RR as this measure is more easily interpretable than OR, as per the Cochrane Handbook for Systematic Reviews of Interventions (Section 6.4.1.2).

COMMENT [Methods] 15- Consider to conduct a subgroup analysis in the LOS outcome according to the study design to solve the heterogeneity. RESPONSE - A subgroup analysis was conducted to explore this (please see Fig S2), however it was not found to account fully for the heterogeneity.

COMMENT [Discussion] 16- "This review explored the evidence available for the impact that ERPs have on outcomes in patients undergoing three common bariatric procedures". You evaluated the impact of ERPs on only one surgery not three, please correct. RESPONSE - The objective of the review was not purely to evaluate the impact but to examine what evidence was available to inform such an evaluation, therefore the confirmation of a lack of evidence for these procedures was part of the findings of the review. We have updated the wording of the opening sentence in the discussion to make this clearer: “This review explored what evidence was available for the impact that ERPs have on outcomes in patients undergoing three common bariatric procedures” [L237]

COMMENT [Discussion] 17- "As only one relevant study was identified for each of LSG and OAGB, meta-analysis was not possible, and results are therefore not discussed for these procedures." State clearly that you exclude the study from your review, as you did not report and data regarding it. RESPONSE - As per the previous comment we have updated the PRISMA flow diagram to make the distinction between studies included in the review versus the analysis clearer. We have also updated the results section to refer to the data supplement where characteristics and outcomes for these studies are presented. [ L162-163]

COMMENT [Discussion] 18- Line 258 "reporting recent clinical and economic evidence related to ERPs in LRYGB, OAGB and LSG within EMEA" You cannot consider this as a strength point as you did not report any data regarding OAGB and LSG. RESPONSE - This sentence has been rephrased to clarify that the review methodology was designed to identify these data as the inability to report analyses is a product of the lack of published data rather than a strength or weakness of the review methodology. [L305]

COMMENT [Discussion] 19- Significant heterogeneity and the relatively small number of studies should be added to the section of limitations. RESPONSE - Thank you for this suggestion, we have added further details of the heterogeneity and the small number of studies to the paragraph discussing evidence gaps. [L297-300]

COMMENT [Discussion] 20- The manuscript needs language editing to eliminate any mistakes. RESPONSE - Thank you, we have performed a full review of the manuscript and updated wording throughout.

Reviewer #3: 

COMMENT - The authors have addressed all my comments/suggestions. RESPONSE - Thank you very much for your review of the revised manuscript

Reviewer #4: 

COMMENT - I have no comments. I accepted author response to manuscripte Enhanced Recovery Programmes Versus Conventional Care in Bariatric Surgery: A Systematic Literature Review and Meta-Analysis. RESPONSE - Thank you very much for your review of the revised manuscript

---

## [Editor Report · Decision Letter 2]

16 Nov 2020

Enhanced Recovery Programmes Versus Conventional Care in Bariatric Surgery: A Systematic Literature Review and Meta-Analysis

PONE-D-20-17131R2

Dear Dr. Steeves,

We’re pleased to inform you that your manuscript has been judged scientifically suitable for publication and will be formally accepted for publication once it meets all outstanding technical requirements.

Kind regards,

Ahmed Negida, MD

Academic Editor

PLOS ONE
---

## [Editor Report · Acceptance letter]

17 Dec 2020

PONE-D-20-17131R2 

Enhanced recovery programmes versus conventional care in bariatric surgery: A systematic literature review and meta-analysis 

Dear Dr. Taylor:

I'm pleased to inform you that your manuscript has been deemed suitable for publication in PLOS ONE. Congratulations! Your manuscript is now with our production department. 

Kind regards, 

on behalf of

Dr. Ahmed Negida 

Academic Editor

PLOS ONE